# Predictors of delayed initiation of breast milk and exclusive breastfeeding in Ethiopia: A multi-level mixed-effect analysis

Gossa Fetene Abebe [1]*, Menen Tilahun[1], Hana Tadesse[1], Abdu Seid[1], Tariku Yigremachew[2], Anteneh Messele Birhanu[2], Desalegn Girma[1]

1 Department of Midwifery, College of Medicine and Health Sciences, Mizan-Tepi University, Mizan-Teferi, Ethiopia, 2 School of Medicine, College of Medicine and Health Sciences, Mizan-Tepi University, Mizan-Teferi, Ethiopia

* Feteneg2119@gmail.com

**Data Availability Statement:** We declared that all the data underlying the results presented in the study are publicly available from the Harvard Dataverse: https://doi.org/10.7910/DVN/YISAFE

## Abstract

### Background

Despite the well-established benefits of early initiation of breastfeeding and exclusive breastfeeding for the first six months to promote optimal neonatal and child health, evidence indicates that in Ethiopia, a significant number of newborns initiate breastfeeding late, do not adhere to exclusive breastfeeding (EBF) for the recommended duration, and instead are fed with bottles.

### Objective

To determine the proportion of delayed initiation of breast milk, exclusive breastfeeding, and its individual and community-level predictors among mothers in Ethiopia.

### Methods

A secondary data analysis was done using the 2019 Ethiopian Mini Demographic Health Survey data. We examined a weighted sample of 2,012 children born within the past 24 months and 623 children aged 0–5 months at the time of the survey. The data analysis was done using STATA version 15. To understand the variation in delayed initiation and exclusive breastfeeding, statistical measures such as the Intraclass correlation coefficient, median odds ratio, and proportional change in variance were calculated. We employed a multilevel mixed-effects logistic regression model to identify predictors for each outcome variable. Statistical significance was determined with a p-value < 0.05.

### Results

The proportion of delayed initiation of breast milk and exclusive breastfeeding were 24.56 and 84.5%, respectively. Women aged 34–49 years old (AOR = 0.33: 95% CI; 0.15–0.72), having a television in the house (AOR = 0.74: 95%CI; 0.33–0.97), delivered by cesarean section (AOR = 3.83: 95% CI; 1.57–9.32), and resided in the Afar regional state (AOR = 1.43: 95%CI; 1.03–12.7) were significantly associated with delayed initiation of breast milk.

**Funding:** The author(s) received no specific funding for this work.

**Competing interests:** The authors have declared that no competing interests exist.

**Abbreviations:** ANC, Antenatal Care; AOR, Adjusted odds ratio; CI, Confidence Interval; SDG, Sustainable Development Goal; WHO, World Health Organization.

On the other hand, attended primary education (AOR = 0.67: 95%CI; 0.35–0.99), secondary education (AOR = 0.34: 95%CI; 0.19–0.53), women whose household headed by male (AOR = 0.68; 95% CI; 0.34–0.97), and rural residents (AOR = 1.98: 95%CI; 1.09–3.43) were significantly associated with exclusive breastfeeding practice.

## Conclusion

Health promotion efforts that encourage timely initation of breast milk and promote EBF, focused on young mothers, those who gave birth through cesarean section, and those residing in urban and the Afar regional state. Furthermore, government health policymakers and relevant stakeholders should consider these identified predictors when revising existing strategies or formulating new policies.

## Introduction

Worldwide, despite several years of focused efforts, neonatal and child mortality and morbidity remain a major public health concern [1]. In 2021, five million children under 5 years of age died globally, which translates to approximately 13,800 under-five deaths occurring every day. These numbers represent an alarming and largely preventable loss of young lives [2].

Breastfeeding is crucial for child health and offers numerous benefits for both children and mothers [3]. Research conducted in lower-income countries has demonstrated that a longer breastfeeding period is linked to improve linear growth, reduced risk of infection, lower neonatal mortality, and decreased likelihood of being underweight in children [4–7]. It also helps in reducing the occurrence of non-specific gastrointestinal tract infections and otitis media. Furthermore, recent studies indicate that breastfeeding for a longer duration offers benefits to mothers as well. These include reduced postpartum blood loss, quicker recovery of the uterus, decreased risk of type I diabetes, and a lower likelihood of becoming overweight in the future [8–10].

Despite the known benefits, many countries still have low rates of appropriate breastfeeding practices [11]. These suboptimal practices result in the loss of approximately 117 million years of life in developing nations [12]. Therefore, to improve the growth, development, and health of children under five, the World Health Organization (WHO) and the United Nations Children's Fund (UNICEF) recommend early initiation of breastfeeding and exclusive breastfeeding for the first six months [11]. Early initiation of breast milk, as defined by the WHO, refers to initiating breastfeeding immediately after birth, preferably within the first hour after delivery [13]. The WHO also recommends exclusive breastfeeding for the first 6 months of life, followed by continued breastfeeding along with appropriate complementary foods for up to 2 years or beyond [14]. However, in developing countries, the rates of delayed initiation of breastfeeding and non-exclusive breastfeeding practices are significant. In Ethiopia, 24.22% of newborns experienced delayed initiation of breastfeeding [15], and 40.1% of children were not exclusively breastfed [16]. To enhance and reinforce early initiation of breast milk and exclusive breastfeeding for the first six months, it is crucial to identify the predictors that act as bottlenecks.

Previous studies conducted in Ethiopia have identified that early initiation of breastfeeding and exclusive breastfeeding practices are challenged by various maternal, child and health service-related predictors. These factors include maternal age [17–19], child age [20–23], marital status [20, 23, 24], employment status [16, 17, 19, 21, 25, 26], religion [27], educational status

of mothers [20, 23, 28, 29], household wealth index [18, 20, 22, 23, 30], ownership of a radio or television [28], sex of household head [31–33], Parity [34–36], sex of child [28, 30, 37, 38], number of family members [16], residence [17, 23, 39, 40], receive infant feeding counselling [17, 21, 22, 25], mothers who had no access to health facility [24], practicing prelacteal feeding [19], institutional delivery [16, 18, 19, 21, 23, 25, 29, 40, 41], giving birth vaginally or by cesarean section [16, 21, 30], history of antenatal care (ANC) [16, 18, 19, 23, 38], postnatal care service utilization [19, 22, 26, 38, 41], and region [16, 30].

However, prior studies employed in Ethiopia had limitations, including specific areas with small sample sizes and a lack of national representativeness. These studies primarily focused on individual-level predictors, neglecting community-level predictors. However, considering contextual predictors is crucial when designing effective service strategies. Currently, there is no research assessing the predictors of initiation of breast milk and exclusive breastfeeding using the recent Ethiopia Mini Demographic and Health Survey (2019 EMDHS). Therefore, this study aims to bridge these gaps by assessing the proportion and predictors, both at the individual and community-levels, of delayed initiation of breast milk and exclusive breastfeeding practices using nationally representative data. The findings will be invaluable in identifying barriers to timely initiation and exclusive breastfeeding practice, informing targeted interventions, and achieving SDG targets related to malnutrition and child mortality reduction by 2030 [42].

## Method and materials

### Study setting, period, and design

The research is conducted using data from the 2019 Ethiopia Mini Demographic and Health Survey (EmDHS). The survey took place from March 21st, 2019 to June 28th, 2019. The EmDHS is a comprehensive study conducted every two to three years in addition to the standard Ethiopia Demographic and Health Surveys (EDHS), which are carried out every five years in nine regions and two city administrations. The EmDHS has been conducted twice, in 2014 and 2019 (Fig 1).

### Eligibility criteria

Our study included all women of reproductive age who had children younger than 24 months and were present in the selected clusters at least one night before the data collection period. Women with missing values were excluded from the analysis. In total, we analyzed data from 2,012 women who had given birth within the last 24 months to examine initiation of breastfeeding. Additionally, we analyzed data from 623 women who had children aged 0–5 months at the time of the survey to assess non-exclusive breastfeeding outcomes.

### Sampling technique

EDHS utilized a two-stage stratified cluster sampling technique. In the first stage, the regions were stratified, and within each region, further stratification was done based on urban and rural areas. In the 2019 EmDHS data, a total of 305 enumeration areas (EAs) were selected in the first stage, with 94 EAs from urban areas and 211 EAs from rural areas. The selection of EAs was based on probability proportional to the size of each EA. In the second stage, households were selected from each EA using a systematic sampling method, proportionally to the size of the EA. The detailed methodology of data collection can be found in the DHS database [43].

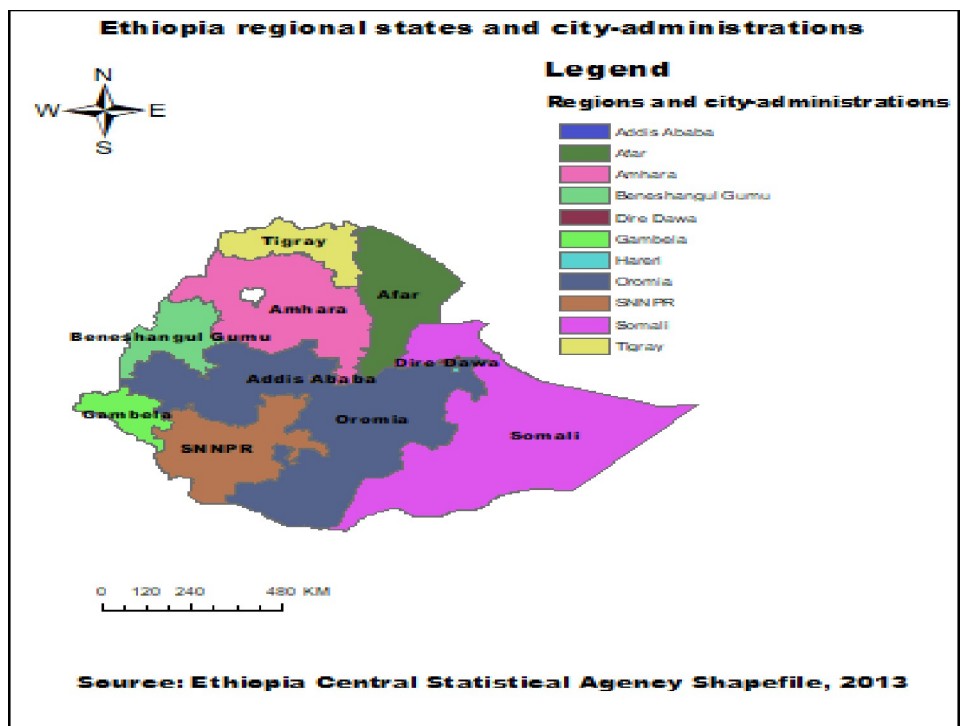

**Fig 1. Regions and city-administrations of the study areas in Ethiopia.**

## Variables of the study

**Response variable.** In our study, the outcome variables were the delayed initiation of breast milk and exclusive breastfeeding practice. Delayed initiation of breast milk was determined if the newborn initiated breastfeeding after one hour of delivery [44]. Exclusive breastfeeding was assessed by whether the infant received only breast milk without any additional food or drink, including water, for the first 6 months of life, except for vitamins, mineral supplements, or medicines. Women who reported feeding their infants (0–5 months) with breast milk only or the allowed liquids mentioned above were labeled as "1," while those who introduced other foods in addition to breast milk were labeled as "0" [44].

**Independent variables.** The study categorized the independent variables into two groups: individual-level variables and community-level variables. Individual-level variables included maternal age, marital status, religion, parity, educational status of mothers, household wealth index, household ownership of a radio, household ownership of a television, sex of household head, sex of child, and the number of family members. On the other hand, community-level variables included the place of residence and region.

## Statistical analysis

The study utilized data from the kid record (KR) file data set and conducted the analysis using STATA version 15. To ensure the survey's representativeness and obtain reliable statistical estimates, the data were weighted for probability sampling and non-response using sample weight (V005). Descriptive statistics were calculated, and the chi-square test was employed to compare the socio-demographic and other profiles of the study participants. The multilevel mixed effects logistic regression model was utilized to assess the predictors of each outcome variable, taking into consideration the hierarchical nature of the EDHS data. Four models were fitted

for each outcome variable in the multilevel logistic regression analysis. The first model aimed to determine the extent of cluster variation in each outcome variable. The second model included individual-level predictors only, while the third model incorporated community-level variables. Finally, the fourth model included both individual and community-level predictors. Both bivariable and multivariable analyses were conducted, with variables having a P-value of ≤ 0.25 in the bivariable multilevel logistic regression analysis being considered as candidates for the multivariable analysis. To declare statistically significant variables, in multilevel multivariable logistic regression model, a P-value of < 0.05 were considered. The variance inflation factor (VIF) was checked to detect the presence of Multicollinearity among covariates. To determine the best fitted model, the log likelihood ratio (LLR) and deviance (-2LLR) value were calculated and a model having high LLR and low deviance value was selected as best fitted model and all interpretations and inferences were made based on this model. To measure the variation of each outcome variable across clusters (EAs), the Intraclass correlation coefficient (ICC), median odds ratio (MOR), and proportional change in variance (PCV) statistics were computed. The ICC was used to determine the variation of each outcome variable within-cluster and between-cluster. The PCV was used to determine the total variation of each outcome variable at the individual- and community-level predictors in each model. The MOR determine the MOR of each outcome variable at the high-risk cluster and low-risk cluster when we randomly select two respondents during data collection from two clusters. To calculate these three measurements, the following formulas are used;

ICC = vi/(vi + π2 /3) ∼ Vi/(Vi+3.29), where Vi = between cluster variances and π2 /3 = within-cluster variance [45].

PCV = (Vi-Vy)/Vi, where Vi = variances of the null model, where Vy = variance of the model with more terms [45].

MOR = exp [√ (2×) Vz ×0.6745] ∼ exp.[0.95√Vz] where Vz = variance at the community level [45].

### Ethical consideration

Since we have used a secondary data analysis that was publicly available from the MEASURE DHS program, ethical approval and participant consent were not required. We requested the DHS program and permission was allowed to download to use the data from: http://www. dhsprogram.com. The requested data were used anonymously and solely for the study's purpose. The full information about the ethical issue was available in the EMDHS-2019 report.

## Results

### The proportion of delayed initiation of breast milk and EBF practice based on the socio-demographic characteristics of study participants

The study found that the overall proportion of delayed initiation of breast milk was 24.56% (95% CI: 21.7–27.6%) (**Fig 2**)**,** while the proportion of exclusive breastfeeding (EBF) practice was 84.5% (95%CI: 79.2–88.6%) (**Fig 3**). Among participants who initiated breastfeeding late, 73.1% were aged 25–34, 94.5% were married, 69.5% lived in rural areas, and 44.5% had no formal education. Mothers from rural areas (88.43%), with no formal education (57.02%), and from the poorest households (22.73%) were less likely to practice EBF for the first six months of the postnatal period (**Table 1**)**.** The Somalia region had the highest proportion of delayed initiation of breast milk (38.8%) (**Fig 4**)**,** while the Amhara region had the highest proportion of EBF practice (95.35%) (**Fig 5**)**.**

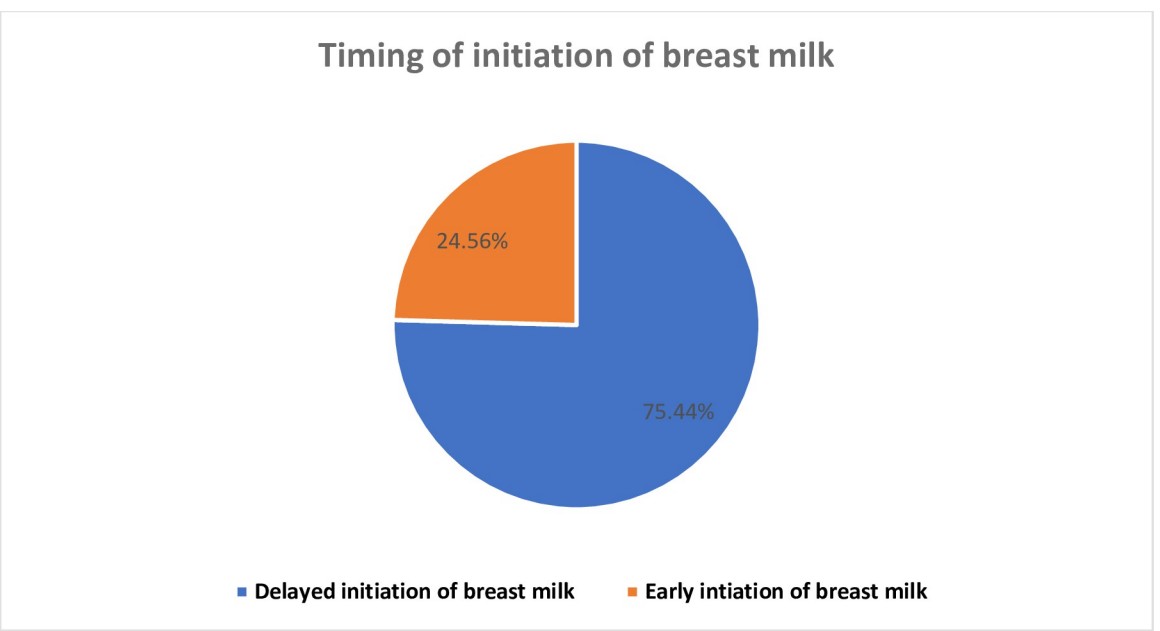

**Fig 2. Timing of initiation of breast milk in Ethiopia using the EMDHS 2019.**

### The proportion of delayed initiation of breast milk and EBF practice based on the different obstetric and reproductive health characteristics of study participants in Ethiopia

Among the participants who experienced delayed initiation of breast milk, approximately 88% gave birth vaginally, 30% did not have any ANC follow-up, 42% had six or more family members, and 19% had a short birth interval. Additionally, the prevalence of EBF practice was highest among women who gave birth vaginally (93.03%) and lowest among women with a short birth interval (17.98%) (**Table 2**).

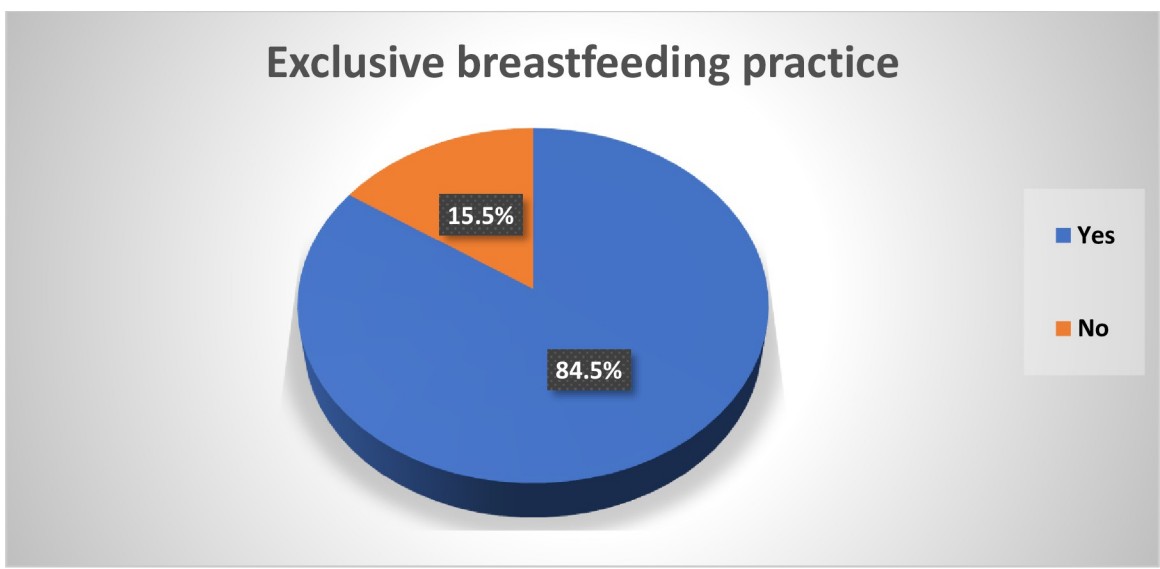

**Fig 3. Exclusive breastfeeding practice in Ethiopia using the EMDHS 2019.**

**Table 1. Proportion of delayed initiation of breast milk, and EBF practice based on the different socio-demographic characteristics of respondents in Ethiopia, 2019.**

| Variable | Category | Delayed initiation of breast milk (n = 2012) | | Exclusive breastfeeding practice for 6 months (n = 623) | |
|---|---|---|---|---|---|
| | | No (%) | Yes (%) | No (%) | Yes (%) |
| Age (in years) | 15–24 | 107 (7.03) | 58 (11.72) | 13 (10.74) | 43 (8.57) |
| | 25–34 | 1,108 (73.02) | 363 (73.14) | 97 (80.17) | 380 (75.70) |
| | ≥ 34 | 303 (19.95) | 73 (14.77) | 11 (9.09) | 79 (15.74) |
| Marital | Married | 1,436 (94.61) | 467 (94.51) | 113 (93.39) | 467 (93.03) |
| | Unmarried | 82 (5.39) | 27 (5.49) | 8 (6.62) | 35 (6.97) |
| Religion | Orthodox | 495 (32.61) | 225 (45.53) | 11 (9.09) | 152 (30.28) |
| | Muslim | 578 (38.08) | 145 (29.27) | 91 (75.21) | 248 (49.40) |
| | Others [c] | 445 (29.31) | 124 (25.21) | 19 (15.70) | 102 (20.32) |
| Residence | Urban | 404 (26.59) | 151 (30.48) | 14 (11.57) | 114 (22.71) |
| | Rural | 1,114 (73.41) | 343 (69.52) | 107 (88.43) | 388 (77.29) |
| Educational status of women | No education | 699 (46.07) | 220 (44.45) | 69 (57.02) | 257 (51.20) |
| | Primary | 613 (40.39) | 202 (40.86) | 37 (30.58) | 171 (34.06) |
| | Secondary/Higher | 205.5 (13.54) | 73 (14.69) | 15 (12.40) | 74 (14.74) |
| Household had radio | No | 1,127 (74.25) | 369 (74.61) | 82 (67.77) | 385 (76.69) |
| | Yes | 391 (25.75) | 125 (25.39) | 39 (32.23) | 117 (23.31) |
| Household had Television | No | 1,240 (81.7) | 408 (82.65) | 108 (89.26) | 420 (83.67) |
| | Yes | 278 (18.3) | 86 (17.35) | 13 (10.74) | 82 (16.33) |
| Household wealth status | Poorest | 431 (67.45) | 208 (32.55) | 50 (22.73) | 170 (77.27) |
| | Poorer | 269 (80.78) | 64 (19.22) | 22 (19.64) | 90 (80.36) |
| | Middle | 210 (74.73) | 71 (25.27) | 12 (14.81) | 69 (85.19) |
| | Richer | 204 (75.84) | 65 (24.16) | 13 (16.67) | 65 (83.33) |
| | Richest | 368 (75.10) | 122 (24.90) | 24 (18.18) | 108 (81.82) |

## Random effect and model comparison for predictors of delayed initiation of breast milk

The ICC in the null model indicates that 23% of the variability in delayed initiation of breast milk can be attributed to differences between clusters or unobserved community-level predictors. Therefore, the multilevel logistic regression model is more appropriate for estimating delayed initiation of breast milk compared to a single-level logistic regression model. Model four, which has a high log-likelihood ratio (-995.6) and low deviance (1,991), is considered the best-fitted model. All interpretations and reports were based on this model. The MOR values in all models are greater than one, indicating variation in delayed initiation of breast milk between clusters. In the null model, the MOR was 3.4, meaning that mothers from clusters with a high proportion of delayed initiation of breast milk were 3.4 times more likely to experience delayed initiation compared to mothers from clusters with a low proportion. The higher PCV value in the fourth model suggests that approximately 57.4% of the difference in delayed initiation of breast milk can be explained by both individual and community-level predictors (**Table 3**).

## Random effect and model comparison for predictors of EBF practice

The ICC in the null model indicates that 48% of the variability in exclusive breastfeeding can be attributed to differences between clusters. Therefore, the multilevel logistic regression model is more suitable for measuring exclusive breastfeeding practice compared to a single-

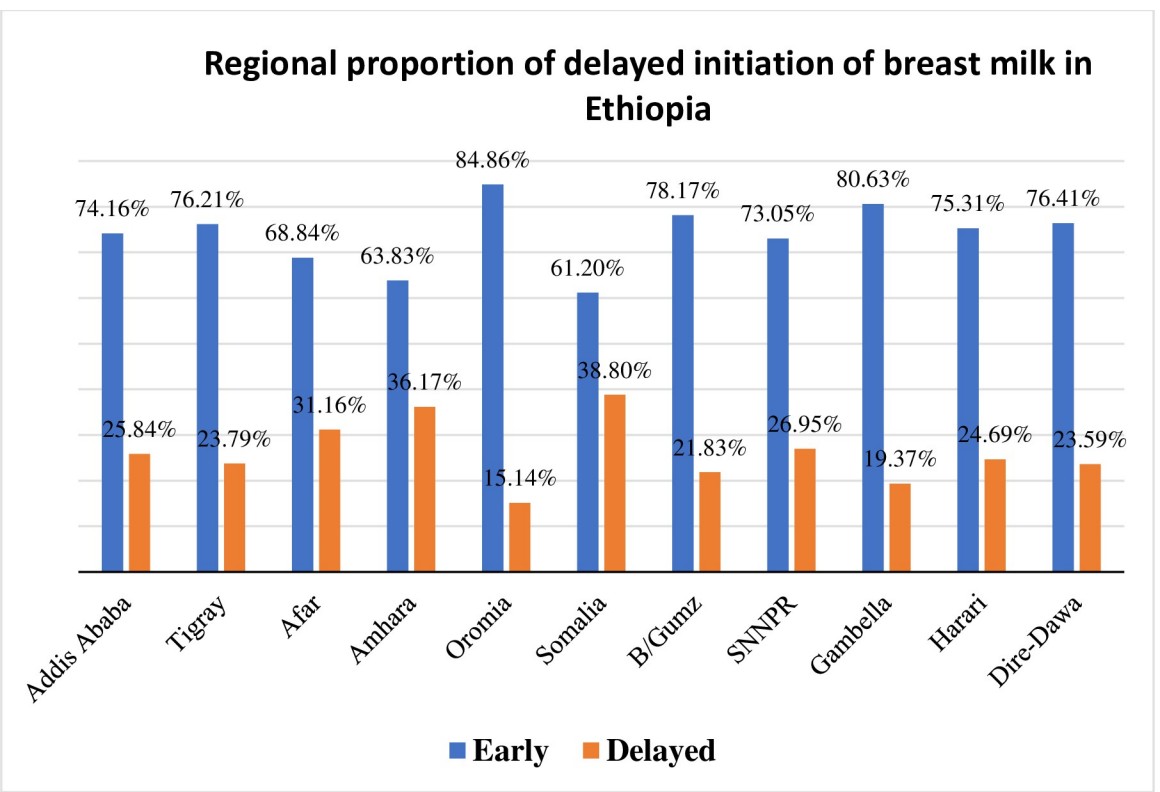

Key; B/Gumz: Benshangul-Gumz region, SNNPR: South Nation Nationality Peoples Region

**Fig 4. Regional proportion of delayed initiation of breast milk in Ethiopia, 2019.**

level logistic regression model. Model four, with a high log-likelihood ratio (-73.34) and low deviance value (146.68), is considered the best-fitted model. All interpretations and reports were based on this model. In all models, the MOR values are greater than one, indicating variation in the practice of exclusive breastfeeding between clusters. In the null model, the MOR was 4.5, meaning that mothers from clusters with a high proportion of exclusive breastfeeding practice were 4.5 times more likely to practice exclusive breastfeeding compared to mothers from clusters with a low proportion. The variations in exclusive breastfeeding practice were best explained by both individual and community-level factors, as evidenced by the higher PCV value in the fourth model. (**Table 4**).

### Predictors of delayed initiation of breast milk in Ethiopia

The final model identified maternal age, household television ownership, mode of delivery, and region as significant predictors of delayed initiation of breast milk in Ethiopia. Mothers aged 34–49 years were 67% less likely to experience delayed initiation of breast milk than mothers aged 15–24 years (AOR = 0.33: 95%CI; 0.15–0.72). Mothers from households with television were 26% less likely to initiate breast milk late than those without television (AOR = 0.74: 95%CI; 0.33–0.97). The likelihood of delayed initiation of breast milk was 3.8 times higher among mothers delivered by cesarean section than those delivered vaginally (AOR = 3.83: 95%CI; 1.57–9.32). Additionally, mothers from the Afar regional state were 1.4

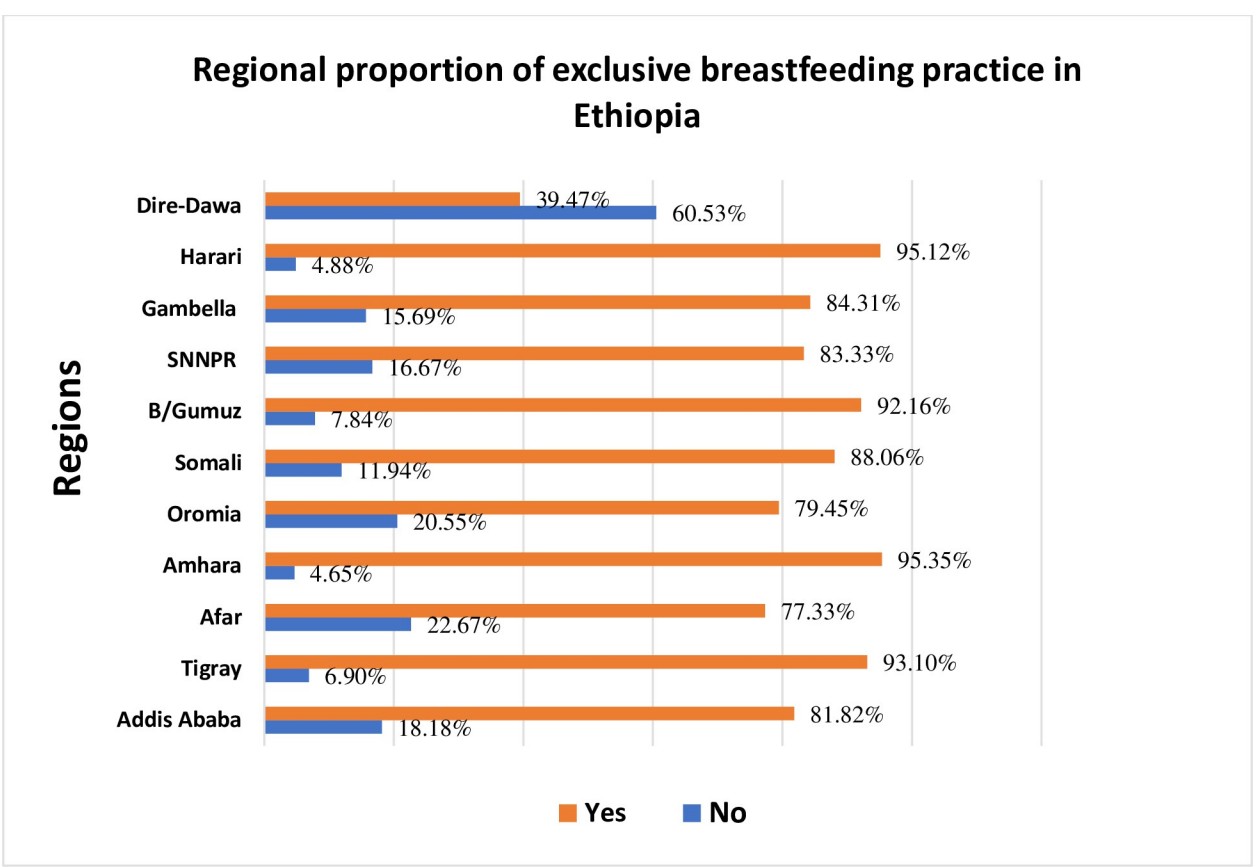

Key; B/Gumz: Benshangul-Gumz region, SNNPR: South Nation Nationality Peoples Region

**Fig 5. Regional proportion of exclusive breastfeeding practice in Ethiopia, 2019.**

times more likely to experience delayed initiation of breast milk than those from Addis Ababa (AOR = 1.43: 95%CI; 1.03–12.7). (**Table 5**).

### Predictors of exclusive breastfeeding practice in Ethiopia

This study found that the educational status of women, female headed household, and being rural residents were significant predictors of exclusive breastfeeding practice in Ethiopia. Women who attended primary education and secondary education or above were 33% and 66% less likely to practice exclusive breastfeeding compared to those who didn't attend formal education, respectively (AOR = 0.67: 95%CI; 0.35–0.99 and AOR = 0.34: 95%CI; 0.19–0.53). Women whose household head was female were 32% less likely to practice exclusive breast-feeding compared to those whose household head was male (AOR = 0.68; 95% CI; 0.34–0.97). Additionally, rural residents were twice as likely to practice exclusive breastfeeding as urban residents (AOR = 1.98: 95%CI; 1.09–3.43) (**Table 6**).

### Discussion

Although the World Health Organization (WHO) recommends that all mothers initiate breast milk within the first hour of birth and exclusively breastfeed for the first six months [46], Ethiopia has not been able to achieve the optimal proportion of delayed initiation of breast milk

**Table 2. Delayed initiation of breast milk and EBF practice based on obstetric and reproductive-health related characteristics of respondents in Ethiopia, 2019.**

| Variable | Category | Delayed initiation of breast milk (n = 2012) | | Exclusive breastfeeding practice for 6 months (n = 623) | |
|---|---|---|---|---|---|
| | | No (%) | Yes (%) | No (%) | Yes (%) |
| Delivered-by cesarean section | No | 1,444 (95.14) | 436 (88.06) | 116 (95.87) | 467 (93.03) |
| | Yes | 74 (4.86) | 59 (11.94) | 5 (4.13) | 35 (6.97) |
| Had ANC visit | No | 296 (29.68) | 101 (29.88) | 25 (28.74) | 122 (35.78) |
| | Yes | 700 (70.32) | 236 (70.12) | 62 (71.26) | 219 (64.22) |
| Household members | ≤ 5 | 768 (50.60) | 287 (58.16) | 64 (52.89) | 227 (45.22) |
| | ≥ 6 | 745 (49.40) | 207 (41.84) | 57 (47.11) | 275 (54.78) |
| Birth interval (in months) | ≤ 23 | 211 (17.54) | 67 (18.90) | 26 (26.53) | 73 (17.98) |
| | ≥ 24 | 994 (82.46) | 289 (81.10) | 72 (73.47) | 333 (82.02) |
| Parity | Primiparous | 142 (26.79) | 297 (20.04) | 22 (18.18) | 95 (18.92) |
| | Multiparous | 244 (46.04) | 711 (47.98) | 62 (51.24) | 248 (49.40) |
| | Grand multiparous | 144 (27.17) | 474 (31.98) | 37 (30.58) | 159 (31.67) |
| Advised danger-signs-during pregnancy | No | 342 (31.20) | 112 (29.29) | 33 (39.29) | 92 (26.14) |
| | Yes | 755 (68.80) | 270 (70.71) | 51 (60.71) | 260 (73.86) |
| Twin | No | 1,486 (97.89) | 487 (98.63) | 116 (95.87) | 467 (93.03) |
| | Yes | 32 (2.11) | 7 (1.37) | 5 (4.13) | 35 (6.97) |
| Place of delivery | Home | 866 (74.14) | 302 (25.86) | 64 (52.89) | 309 (61.55) |
| | Health facility | 616 (72.99) | 228 (27.01) | 57 (47.11) | 193 (38.45) |

and exclusive breastfeeding practice [16]. Consequently, this study aims to offer up-to-date information on the proportion and determinants of delayed initiation of breast milk and exclusive breastfeeding practices in Ethiopia. The study utilized data from the 2019 Ethiopia Mini Demographic and Health Survey.

In this study, the proportion of delayed initiation of breast milk was 24.56% (95% CI: 21.7–27.6%). The result is in line with the studies done in Ethiopia, for example, a secondary analysis using the 2016 EDHS data done by Ahmed et. al. (24.3%) [16], Debre Tabor (23.2%) [47], and Dembecha district (26.9%) [48].

However, the result is lower than studies done in Ethiopia, for example, Dire Dawa city (29.1%) [34], Arsi Zone (32.7%) [29], Mizan-Aman town (35.5%) [49], and Axum town (58.4%) [37]. The divergence in results could be due to variations in the study period [29, 37], the implementation and reinforcement of the Health Extension Program (HEP) and women development army, and the enhancement of healthcare accessibility in the country.

**Table 3. Random effect and model comparison for predictors of delayed initiation of breast milk among mothers who had children less than two years in Ethiopia, 2019.**

| Parameter | Null (model I) | Model II | Model III | Model VI |
|---|---|---|---|---|
| ICC | 23% | 23.8% | 18% | 16% |
| Variance | 1.76 (1.02–1.98) | 0.98 (0.62–1.57) | 0.73 (0.46–1.15) | 0.75 (0.47–1.20) |
| MOR | 3.4 | 2.5 | 2.2 | 2.23 |
| PCV | Reference | 44.3% | 58.5% | 57.4% |
| Model fitness | | | | |
| Log likelihood ratio (LLR) | -1032.1 | -1007.3 | -1020.3 | -995.6 |
| Deviance (-2LLR) | 2,064.2 | 2,014.6 | 2,040.6 | 1,991 |

ICC: Intraclass correlation coefficient, MOR: Median Odds ratio, PCV: proportional change in variance

**Table 4. Random effect and model comparison for determinants of exclusive breastfeeding among mothers who had children less than six months in Ethiopia, 2019.**

| Parameter | Null (model I) | Model II | Model III | Model VI |
|---|---|---|---|---|
| ICC | 48% | 38.6% | 45.3% | 40.9% |
| Variance | 3.05 (1.6–5.7) | 2.1 (0.6–6.8) | 1.56 (1.2–1.8) | 1.24 (0.6–2.2) |
| MOR | 4.51 | 3.74 | 5.05 | 2.88 |
| PCV | Reference | 31.1% | 48.8% | 59.3% |
| Model fitness | | | | |
| Log likelihood ratio (LLR) | -209.06 | -77.42 | -201.09 | -73.34 |
| Deviance (-2LLR) | 418.12 | 154.84 | 402.18 | 146.68 |

ICC: Intraclass correlation coefficient, MOR: Median Odds ratio, PCV: proportional change in variance

**Table 5. Multilevel mixed-effect logistic regression analysis to determine the predictors of the delayed initiation of breast milk in Ethiopia, 2019.**

| Variable | Category | Null (Model I) | Model II AOR (95%CI) | Model III AOR (95%CI) | Model VI AOR (95%CI) |
|---|---|---|---|---|---|
| Maternal age (in years) | 15–24 | - | 1 | - | 1 |
| | 25–34 | - | 0.56 (0.27, 1.17) | - | 0.55 (0.27, 1.15) |
| | 35–49 | - | 0.32 (0.15, 0.71) | - | **0.33 (0.15, 0.72)** * |
| Marital status | Married | - | 1 | - | 1 |
| | Unmarried | - | 0 .65 (0.25, 1.68) | - | 0.71 (0.28, 1.82) |
| Educational status | No education | - | 1 | - | 1 |
| | Primary | - | 0.87 (0.57, 1.33) | - | 0.94 (0.60, 1.47) |
| | Secondary/tertiary | - | 0.87 (0.47, 1.63) | - | 0.97 (0.52, 1.83) |
| Household had television | No | - | 1 | - | 1 |
| | Yes | - | 0.72 (0.34, 1.52) | - | **0.74 (0.33, 0.97)** * |
| Sex of child | Male | - | 1 | - | 1 |
| | Female | - | 1.05 (0.68, 1.62) | - | 1.05 (0.69, 1.60) |
| Delivered by cesarean section | No | - | 1 | - | 1 |
| | Yes | - | 3.92 (1.61, 9.55) | - | **3.83 (1.57, 9.32)** * |
| Community-level variables | | | | | |
| Residence | Urban | - | - | 1 | 1 |
| | Rural | - | - | 0.92 (0.57, 1.49) | 0.88 (0.43, 1.82) |
| Region | Addis Ababa | - | - | 1 | 1 |
| | Tigray | - | - | 0.89 (0.42, 1.91) | 0.95 (0.36, 2.55) |
| | Afar | - | - | 1.47 (0.64, 3.37) | **1.43 (1.03, 12.7)** * |
| | Amhara | - | - | 1.47 (0 .71, 3.04) | 1.65 (0.63, 4.33) |
| | Oromia | | - | 0.48 (0.22, 1.02) | 0.53 (0.19, 1.49) |
| | Somalia | | - | 1.80 (0.79, 4.08) | 1.82 (0.87, 5.61) |
| | B/Gumz | | - | 0.76 (0.35, 1.66) | 0.76 (0.26, 2.21) |
| | SNNPR | | - | 1.15 (0.56, 2.34) | 1.23 (0.43, 3.50) |
| | Gambella | | - | 0.67 (0.30, 1.52) | 0.66 (0.22, 2.04) |
| | Harari | | - | 0.95 (0.43, 2.07) | 1.02 (0.41, 2.54) |
| | Dire Dawa | | - | 0.89 (0.41, 1.91) | 0.82 (0.34, 1.97) |

*p-value < 0.05, AOR: Adjusted odds ratio, CI: Confidence interval, 1: Reference, B/Gumiz: Benshangul-Gumz region, SNNPR: South Nation Nationality people's region

**Table 6. Multilevel mixed-effect logistic regression analysis to identify the predictors of the EBF practice in Ethiopia, 2019.**

| Variable | Category | Null (Model I) | Model II AOR (95%CI) | Model III AOR (95%CI) | Model VI AOR (95%CI) |
|---|---|---|---|---|---|
| Maternal age (in years) | 15–24 | - | 1 | - | 1 |
| | 25–34 | - | 0.53 (0.19, 1.19) | - | 0.05 (0.01, 1.31) |
| | 35–49 | - | 0.6 (0.13, 2.49) | - | 0.07 (0.07, 3.7) |
| Religion grouped | Orthodox | - | 1 | - | 1 |
| | Muslim | | 0.72 (0.41, 0.97) | | 0.63 (0.46, 1.23) |
| | Others[C] | - | 0.62 (0.31, 0.86) | - | 0.85 (0.42, 1.98) |
| Educational status | No education | - | 1 | - | 1 |
| | Primary | - | 0.45 (0.3, 0.56) | - | **0.67 (0.35, 0.99)** * |
| | Secondary/tertiary | - | 0.33 (0.24, 0.49) | - | **0.34 (0.19, 0.53)** * |
| Household wealth index | Poorest | - | 1 | - | 1 |
| | Poorer | - | 0.56 (0.08, 3.84) | - | 0.57 (0.05, 6.18) |
| | Middle | - | 1.06 (0.03, 14.32) | - | 1.0(0.01, 7.3) |
| | Rich | - | 0.31 (0.02, 4.76) | - | 0.31(0.02, 6.08) |
| | Richest | - | 1.81 (0.12, 11.74) | - | 2.56 (0.05, 8.2) |
| Number of Under-five child | ≤ 2 | - | 1 | - | 1 |
| | ≥ 3 | - | 0.52 (0.12, 2.48) | - | 0.46 (0.08, 2.75) |
| Birth interval (in months) | ≤ 23 | - | 1 | - | 1 |
| | ≥ 24 | - | 1.41 (0.36, 5.57) | - | 1.35 (0.29, 6.32) |
| Household had television | No | - | 1 | - | 1 |
| | Yes | - | 0.47 (0.10, 2.12) | - | 0.73 (0.02, 12) |
| Household had Radio | No | - | 1 | - | 1 |
| | Yes | - | 0.59 (0.03, 11.52) | - | 0.53 (0.09, 3.0) |
| Sex of household head | Male | - | 1 | - | 1 |
| | Female | - | 0.13 (0.03, 0.63) | - | **0.68 (0.34, 0.97)** * |
| Had ANC visit | No | - | 1 | - | 1 |
| | Yes | - | 0.94 (0.09, 8.91) | - | 0.91 (0.07, 12.21) |
| Place of delivery | Home | - | 1 | - | 1 |
| | Health facility | - | 0.38 (0.08, 1.79) | - | 0.46 (0.08, 2.48) |
| Delayed initiation of breast milk | No | - | 1 | - | 1 |
| | Yes | - | 1.78 (0.51, 6.17) | - | 1.92 (0.45, 8.18) |
| Current child age (in month) | 0–2 | - | 1 | - | 1 |
| | 3–5 | - | 0.26 (0.05, 1.33) | - | 0.21 (0.03, 1.58) |
| Community-level variables | | | | | |
| Residence | Urban | - | - | 1 | 1 |
| | Rural | - | - | 0.59 (0.12, 3.36) | **1.98 (1.09, 3.43)** * |
| Region | Addis Ababa | - | - | 1 | 1 |
| | Tigray | - | - | 4.03 (0.26, 18) | 2.62 (0.25, 11.53) |
| | Afar | - | - | 0.58 (0.04, 7.43) | 0.38 (0.05, 3.16) |
| | Amhara | - | - | 5.86 (0.21, 16.3) | 3.91 (0.27, 7.91) |
| | Oromia | | - | 0.84 (0.05, 13.59) | 0.52 (0.06, 4.36) |
| | Somalia | | - | 1.75 (0.13, 8.74) | 1.13 (0.13, 9.83) |
| | B/Gumz | | - | 4.46 (0.35, 15.54) | 3.02 (0.32, 8.91) |
| | SNNPR | | - | 1.89 (0.14, 12.23) | 1.19 (0.14, 9.92) |
| | Gambella | | - | 2.32 (0.18, 17.53) | 1.80 (0.16, 20.07) |
| | Harari | | - | 4.41 (0.29, 7.85) | 3.14 (0.26, 17.79) |
| | Dire Dawa | | - | 0.08 (0.01, 1.17) | 0.06 (0.01, 1.75) |

*p-value < 0.05, AOR: Adjusted odds ratio, CI: Confidence interval, 1: Reference

[C] Protestant, catholic or traditional religion follower, SNNPR: South Nation Nationality people's region, B/Gumiz: Benshangul-Gumz region

Moreover, the delayed initiation of breast milk in this study is higher than studies conducted in different parts of Ethiopia like Bedessa district (18.8%) [41], Gunchire town (19.5%) [50], Mekelle town (22.1%) [51] and Western Ethiopia (11.5%) [52]. This variation may be attributed to differences in the study settings (previous studies were conducted in a single district or town with a small sample size) and study populations; most of the participants in previous studies were urban dwellers, while 72.4% and 79.5% of the mothers who participated in the initiation of breast milk and EBF practice in this study were rural residents. So, the higher proportion of delayed initiation of breast milk reported in this study may be attributed to the fact that women living in rural areas [53] are more prone to delivering outside of health facilities (i.e., at home) without the aid of healthcare professionals. Additionally, these mothers may lack information on the timely initiation of breast milk, leading to a failure to initiate breast milk early.

The study also found that the proportion of EBF practice in Ethiopia was 84.5% (95%CI: 79.2–88.6%). This aligns with similar studies conducted in Ethiopia, such as those in the Afar region (81%) [54], and Ambo district (82%) [55]. It is also consistent to study employed in Northern Gahana [56]. Notably, this study reported a higher EBF proportion compared to studies conducted in Ethiopia, including Ahmed et al. (59.9%) [16], EDHS 2016 (58%) [57], Somali region (52%) [58], Mecha district (47.13%) [59], and Axum town (40.9%) [37]. The variation observed could be attributed to differences in the study population, timeframes, and settings. In conclusion, this study demonstrates significant progress in the practice of exclusive breastfeeding (EBF) in Ethiopia.

In line with prior research conducted in Ethiopia [36, 60], the findings of this study indicate that mothers aged 34–49 years were less likely to encounter delayed initiation of breastfeeding in comparison to mothers aged 15–24 years. One possible explanation for this trend is that as maternal age increases, so does the mother's experience in managing infants, which can facilitate the timely initiation of breastfeeding within one hour of birth.

Mothers residing in households with access to television were discovered to have a lower likelihood of delaying the initiation of breast milk compared to those without television. This finding aligns with a study conducted in Northwest Ethiopia [48]. One possible explanation for this correlation could be the acquisition of information through television channels, which may enhance the knowledge and practices of mothers regarding breastfeeding. As a result, they are more likely to initiate breast milk in a timely manner.

In agreement with previous studies done in Ethiopia [16, 30, 36], mothers who undergo cesarean section have a higher likelihood of delayed initiation of breast milk compared to those who deliver vaginally. This may be due to the longer time of the procedure, postoperative pain, anesthesia effects, fatigue, and delayed mother-baby contact during postoperative care.

Regional variations in the timely initiation of breast milk were observed in Ethiopia. Mothers residing in the Afar regional state exhibited a higher likelihood of delayed initiation compared to mothers in Addis Ababa, the country's capital city. This disparity can be attributed to differences in the accessibility and availability of healthcare facilities [61]; participants from Addis Ababa are more likely to give birth at healthcare facilities under the guidance of healthcare professionals, providing them with better information regarding the advantages of initiating breastfeeding in a timely manner.

The practice of EBF was adversely affected by the educational status of women. Women who received formal education were less likely to engage in EBF compared to those who did not attend any formal education. This observation aligns with previous research conducted in Ethiopia [19, 57, 62]. One possible explanation for this phenomenon is that when women have higher levels of education, the likelihood of pursuing employment increases, which

compromises their ability to stay at home and practice EBF. Additionally, women may also be influenced by media advertisements promoting milk substitutes.

Mothers who assumed the role of household head were found to be less likely to engage in EBF compared to those whose household head was male. This finding is supported with the previous studies done in Ethiopia, for instance, Woldia town [31], and Hossana town [33]. This could be attributed to the impact of the increased workload on mothers, resulting in reduced time available for feeding the child. In families where such circumstances prevail, mothers may find themselves burdened with responsibilities extending beyond the traditional roles of leading a family, fulfilling social obligations, managing economic matters, and handling miscellaneous tasks. This additional workload can significantly limit the time mothers have to dedicate to feeding their child.

Moreover, rural residents exhibited a twofold higher likelihood of engaging in EBF compared to their urban counterparts. This finding is consistent with previous research conducted in Ethiopia [32], and Malaysia [63]. There are two possible reasons for this phenomenon. Firstly, women residing in urban areas are more likely to have received higher levels of education [64] and have greater access to diverse employment opportunities, which can limit the amount of time they can spend with their infants, thereby compromising the practice of exclusive breastfeeding. Alternatively, it could be attributed to the fact that urban mothers have greater access to alternative infant feeding options compared to their rural counterparts.

## Policy and practice implications

This study adds to the existing knowledge on the delayed initiation of breastfeeding and the practice of EBF, as well as the factors that hinder mothers from adopting these practices. Policymakers and planners can utilize these identified determinants to improve breastfeeding practices, thereby reducing complications in neonates and children associated with inadequate breastfeeding. The Ethiopian Federal Ministry of Health, in collaboration with NGOs, should prioritize support for older mothers, those in rural areas, households led by females, individuals with limited formal education, women who delivered via cesarean section, and those residing in the Afar region. This focus will help maximize early initiation of breastfeeding and the practice of EBF.

## Strength and limitation of the study

The study had strengths in utilizing nationally representative data, ensuring a large sample size, high response rate, and high-quality data. These minimized biases related to sampling and measurement. Additionally, an appropriate statistical approach, multilevel mixed-effect analysis, accurately estimated the impact of cluster effects on delayed initiation of breastfeeding and the practice of exclusive breastfeeding. However, limitations include the inability to establish a definitive cause-effect relationship due to the cross-sectional design, potential recall bias, and the omission of important variables like maternal and infant problems after delivery, maternal knowledge, distance to health facilities, and pregnancy intention, which could potentially influence the outcomes under investigation.

## Conclusion

In summary, the practice of EBF in Ethiopia has shown promising progress. However, the initiation of breast milk has not undergone significant change and remains a concern. Delayed initiation of breast milk is significantly associated with maternal age, access to Television, delivered by cesarean section, and lived in Afar region. Whereas, EBF practice is correlated with educational status, female headed household, and being rural residents. Therefore,

enhancing female education and economic transitions with special consideration given to rural, and Afar regional state residents could maximize the early initiation of breast milk and EBF practice. Counseling towards the benefits of early initiation of breast milk and EBF practice should be delivered much strengthened to maximize full utilization of appropriate breast-feeding practices. Furthermore, to increase the early initiation of breast milk and EBF practice, the identified predictors should be underscored when designing new policies or updating policies and strategies on appropriate breastfeeding practice to step-up its full utilization, which in turn help to achieve SDG targets related to malnutrition and child mortality reduction by 2030.

## Acknowledgments

The authors acknowledge the Demographic and Health Surveys center for allowing and permitting to access the data set.

## Author Contributions

**Conceptualization:** Gossa Fetene Abebe, Menen Tilahun, Hana Tadesse, Abdu Seid, Tariku Yigremachew, Anteneh Messele Birhanu, Desalegn Girma.

**Data curation:** Gossa Fetene Abebe, Menen Tilahun, Hana Tadesse, Abdu Seid, Tariku Yigremachew, Anteneh Messele Birhanu, Desalegn Girma.

**Formal analysis:** Gossa Fetene Abebe, Hana Tadesse, Abdu Seid, Anteneh Messele Birhanu, Desalegn Girma.

**Methodology:** Gossa Fetene Abebe, Anteneh Messele Birhanu, Desalegn Girma.

**Resources:** Menen Tilahun.

**Software:** Gossa Fetene Abebe, Menen Tilahun, Abdu Seid.

**Validation:** Gossa Fetene Abebe, Menen Tilahun, Hana Tadesse.

**Visualization:** Menen Tilahun, Tariku Yigremachew, Desalegn Girma.

**Writing – original draft:** Gossa Fetene Abebe.

**Writing – review & editing:** Gossa Fetene Abebe, Menen Tilahun, Hana Tadesse, Abdu Seid, Tariku Yigremachew, Anteneh Messele Birhanu, Desalegn Girma.

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
