## [Decision Letter · Decision Letter 0]

26 Feb 2024

PONE-D-23-34507Predictors of Delayed Initiation of Breast Milk and Exclusive Breastfeeding in Ethiopia: A Multi-level Mixed-effect AnalysisPLOS ONE

Dear Dr. Gossa,

Thank you for submitting your manuscript to PLOS ONE. After careful consideration, we feel that it has merit but does not fully meet PLOS ONE’s publication criteria as it currently stands. Therefore, we invite you to submit a revised version of the manuscript that addresses the points raised during the review process.

We look forward to receiving your revised manuscript.

Kind regards,

Kahsu Gebrekidan

Academic Editor

PLOS ONE

Journal Requirements:

4. We note that Figure 1 in your submission contain map/satellite image which may be copyrighted. All PLOS content is published under the Creative Commons Attribution License (CC BY 4.0), which means that the manuscript, images, and Supporting Information files will be freely available online, and any third party is permitted to access, download, copy, distribute, and use these materials in any way, even commercially, with proper attribution. For these reasons, we cannot publish previously copyrighted maps or satellite images created using proprietary data, such as Google software (Google Maps, Street View, and Earth). For more information, see our copyright guidelines: http://journals.plos.org/plosone/s/licenses-and-copyright.

Reviewers' comments:

Reviewer's Responses to Questions

**Comments to the Author**

1. Is the manuscript technically sound, and do the data support the conclusions?

Reviewer #1: Yes

Reviewer #2: Yes

2. Has the statistical analysis been performed appropriately and rigorously? 

Reviewer #1: Yes

Reviewer #2: Yes

3. Have the authors made all data underlying the findings in their manuscript fully available?

Reviewer #1: Yes

Reviewer #2: Yes

4. Is the manuscript presented in an intelligible fashion and written in standard English?

Reviewer #1: Yes

Reviewer #2: Yes

5. Review Comments to the Author

Reviewer #1: Reviewer Comments to the Author

Dear PLOS One team of editorials, thank you for giving me the chance to review the manuscript entitled " Predictors of Delayed Initiation of Breast Milk and Exclusive Breastfeeding in Ethiopia: A Multi-level Mixed-effect Analysis”

This study gives very important results regarding delayed initiation of breast milk and exclusive breastfeeding. However, in a few areas, here are my comments.

1. While reading the abstract …Which software you used for data analysis is not clear ....it must be included in the abstract…. The abstract is many-worded try to make it short, catchier, and reader-friendly... conclusion does not match with the objective.

2. WHO recommends exclusive breastfeeding for the first 6 months of life, but in your study, you include ages 0-5 ...why??

3. Keywords are not matched with the title

4. In the background major consequences or effects of delayed initiation of feeding or not doing exclusive breastfeeding are not discussed.

5. There are dissimilarities in study objectives: in abstract and introduction

6. Revisit the numbers, totals, and the statistics in general

7. Insufficient citation, particularly in discussion for safe interpretation

8. It is not mandatory to report the strengths of your study.

9. What are the measures taken to control the confounders?

10. Use correct tense, grammar, sentence, spelling, paraphrase, consistency…check it.

Reviewer #2: Sampling was done about 5 years ago. Does this data look old? Considering the importance of publishing DHS data, how do the authors justify the reason for this delay?

One of the cases that seems to be necessary to be considered in the criteria for selecting people is the cases where it is not possible to start early feeding with breast milk after giving birth due to maternal and infant problems, which is a completely scientific and logical separation. When the results of the delay in the initiation of breastfeeding for this high-risk mother or infant are combined with the delay in the onset of breastfeeding (newborns or mothers who did not have any problems after delivery), it leads to a higher indication of the delay in the initiation of breastfeeding. How did the researchers deal with this problem?

6. PLOS authors have the option to publish the peer review history of their article (what does this mean?). If published, this will include your full peer review and any attached files.

Reviewer #1: No

Reviewer #2: No

---

## [Author Response · Author response to Decision Letter 0]

28 Feb 2024

Author Response to Editor and Reviewers

Dear Editor and Reviewers,

We would like to express our gratitude for your email dated 26 Feb 2024, which included the insightful comments from the editor and reviewers. We, the authors, sincerely appreciate the valuable and constructive review that has greatly contributed to the improvement of our paper titled "Predictors of Delayed Initiation of Breast Milk and Exclusive Breastfeeding in Ethiopia: A Multi-level Mixed-effect Analysis". We have thoroughly reviewed the editor's and reviewers' comments and have made the necessary revisions to the manuscript accordingly. Our responses to the editor and reviewers' comments are provided in a point-by-point manner using the Author's response to reviewer form. If you have any further concerns or suggestions, we are more than willing to address them.

Best regards,

Version 1: PONE-D-23-34507

Date: 2/27/2024

Academic editor comments and respective author’s response 

Editor comment 1: Please ensure that your manuscript meets PLOS ONE's style requirements, including those for file naming. The PLOS ONE style templates can be found at https://journals.plos.org/plosone/s/file?id=wjVg/PLOSOne_formatting_sample_main_body.pdf and https://journals.plos.org/plosone/s/file?id=ba62/PLOSOne_ formatting_ sample_ title_authors_affiliations.pdf

Authors Response: Thanks very much for this comment. The whole part of the manuscript has been documented as per the PLOS ONE style templates. 

Editor comment 2: Please provide additional details regarding participant consent. In the ethics statement in the Methods and online submission information, please ensure that you have specified (1) whether consent was informed and (2) what type you obtained (for instance, written or verbal, and if verbal, how it was documented and witnessed). If your study included minors, state whether you obtained consent from parents or guardians. If the need for consent was waived by the ethics committee, please include this information.

Authors Response: Thanks very much for this comment. We used an aggregated publicly available secondary data taken from the Measure DHS program (http://www.dhsprogram.com), which does not include any personal identifiers that could be linked to the study participants (See page 9, under the sub-heading ‘‘Ethical consideration’’). 

Editor comment 3: Note from Emily Chenette, Editor in Chief of PLOS ONE, and Iain Hrynaszkiewicz, Director of Open Research Solutions at PLOS: Did you know that depositing data in a repository is associated with up to a 25% citation advantage……

Authors Response: Thanks very much for this supportive comment. We updated the data availability statement as ‘‘We declared that all the data underlying the results presented in the study are publicly available from the Harvard Dataverse; https://doi.org/10.7910/DVN/YISAFE’’.

Editor comment 4: We note that Figure 1 in your submission contain map/satellite image which may be copyrighted. All PLOS content is published under the Creative Commons Attribution License (CC BY 4.0), which means that the manuscript, images, and Supporting Information files will be freely available online, and any third party is permitted to access, download, copy, distribute, and use these materials in any way, even commercially, with proper attribution…… 

Authors Response: Thanks very much for this insightful comment. We, the authors, assure that the map used in figure 1 are freely available at: https://africaopendata.org/dataset/ethiopia-shapefiles.

Editor comment 5: Please review your reference list to ensure that it is complete and correct…..

Authors Response: Thanks very much for this constructive comment. We, the authors, reviewed, checked, and ensured that all the reference lists were complete and correct. 

Reviewer #1 comments and an author’s response

Dear PLOS One team of editorials, thank you for giving me the chance to review the manuscript entitled " Predictors of Delayed Initiation of Breast Milk and Exclusive Breastfeeding in Ethiopia: A Multi-level Mixed-effect Analysis”

This study gives very important results regarding delayed initiation of breast milk and exclusive breastfeeding. However, in a few areas, here are my comments.

Comment #1: While reading the abstract …Which software you used for data analysis is not clear ....it must be included in the abstract…. The abstract is many-worded try to make it short, catchier, and reader-friendly... conclusion does not match with the objective.

Authors Response: Thanks very much, dear reviewer, for these insightful comments. We accept the comments and corrections have been made accordingly (See pages 2 to 3 under the heading ‘‘Abstract’’).

Comment #2: WHO recommends exclusive breastfeeding for the first 6 months of life, but in your study, you include ages 0-5 ...why??

Authors Response: Thanks very much, dear reviewer, for this critical comment. The WHO define exclusive breast feeding as the infant received only breast milk without any additional food or drink, including water, for the first 6 months of life (0-5 months), except for vitamins, mineral supplements, or medicines (WHO, 2023). 

WHO: Exclusive breastfeeding for optimal growth, development and health of infants; Updated on 9 August 2023. Available at; https://www.who.int/tools/elena/interventions/exclusive-breastfeeding#:∼:text=Exclusive%20breastfeeding%20means%20that%20the,of%20vitamins%2C%20minerals%20or%20medicines. 2023.

Comment #3: Keywords are not matched with the title

Authors Response: Thanks very much, dear reviewer, for this insightful comment. We accept the comment and correction has been made accordingly (See page 3, line 9). 

Comment #4: In the background major consequences or effects of delayed initiation of feeding or not doing exclusive breastfeeding are not discussed.

Authors Response: Thanks very much for this constructive comment. The major consequences or effects of delayed initiation of feeding or not doing exclusive breastfeeding are discussed under the introduction section, paragraph two (See page 3, lines 15–22).

Comment #5: There are dissimilarities in study objectives: in abstract and introduction

Authors Response: Thank you very much for your insightful comment. We greatly appreciate it. The comment has been accepted, and revisions have been made to ensure that the objectives in the abstract section are aligned with the introduction (See page 2, lines 7-8, and page 5, lines 9-11).

Comment #6: Revisit the numbers, totals, and the statistics in general.

Authors Response: Thanks very much, dear reviewer, for this insightful suggestion. We, the authors, assured that all the numbers and percentages have been checked and all are correct. 

Comment #7: Insufficient citation, particularly in discussion for safe interpretation.

Authors Response: Thanks very much, dear reviewer, for this insightful comment. We, the authors, accept the comment and revised the citation in the discussion section accordingly. 

Comment #8: It is not mandatory to report the strengths of your study.

Authors Response: Thank you very much, dear reviewer, for your insightful suggestion. We, the authors, genuinely appreciate your concern. However, we are keen on providing a comprehensive explanation of both the strengths and limitations of the study and addressed under the section ‘‘strength and limitations of the study’’. 

Comment #9: What are the measures taken to control the confounders?

Comment #10: Use correct tense, grammar, sentence, spelling, paraphrase, consistency. check it.

Authors Response: Thanks very much, dear reviewer, for this insightful suggestion. We, the authors, assured that all things included in the document have been checked and correct. 

Reviewer #2 comments and authors’ response

Comment #1: Sampling was done about 5 years ago. Does this data look old? Considering the importance of publishing DHS data, how do the authors justify the reason for this delay?

Authors Response: Thank you, dear reviewer, for your questions. We, the authors, confirm that the data used in our study was obtained from the most recent Ethiopia Mini Demographic and Health Survey 2019. This survey collected information from respondents within the five years prior to the survey period. The survey took place from March 21st, 2019 to June 28th, 2019 (EPHI, 2019). 

Ethiopia Public Health Institute, ICF: Ethiopia mini demographic and health survey 2019: key indicators. Rockville, Maryland, USA: EPHI and ICF 2019.

Comment #2: One of the cases that seems to be necessary to be considered in the criteria for selecting people is the cases where it is not possible to start early feeding with breast milk after giving birth due to maternal and infant problems, which is a completely scientific and logical separation. When the results of the delay in the initiation of breastfeeding for this high-risk mother or infant are combined with the delay in the onset of breastfeeding (newborns or mothers who did not have any problems after delivery), it leads to a higher indication of the delay in the initiation of breastfeeding. How did the researchers deal with this problem?

Authors Response: Thank you, dear reviewer, for your insightful comment and question. We, the authors, appreciate your concern. As we utilized secondary data, the variables regarding maternal and infant problems were not specifically documented, leading to our inability to classify which women or newborns experienced problems after delivery. We acknowledge this as a limitation of our study and have included it in the "Strengths and Limitations" section (See page 23, line 7).

---

## [Decision Letter · Decision Letter 1]

11 Mar 2024

Predictors of Delayed Initiation of Breast Milk and Exclusive Breastfeeding in Ethiopia: A Multi-level Mixed-effect Analysis

PONE-D-23-34507R1

Dear Mr Gossa,

We’re pleased to inform you that your manuscript has been judged scientifically suitable for publication and will be formally accepted for publication once it meets all outstanding technical requirements.

Kind regards,

Kahsu Gebrekidan

Academic Editor

PLOS ONE

Additional Editor Comments (optional):

Reviewers' comments:

Reviewer's Responses to Questions

**Comments to the Author**

1. If the authors have adequately addressed your comments raised in a previous round of review and you feel that this manuscript is now acceptable for publication, you may indicate that here to bypass the “Comments to the Author” section, enter your conflict of interest statement in the “Confidential to Editor” section, and submit your "Accept" recommendation.

Reviewer #1: All comments have been addressed

Reviewer #2: All comments have been addressed

2. Is the manuscript technically sound, and do the data support the conclusions?

Reviewer #1: Yes

Reviewer #2: Yes

3. Has the statistical analysis been performed appropriately and rigorously? 

Reviewer #1: Yes

Reviewer #2: Yes

4. Have the authors made all data underlying the findings in their manuscript fully available?

Reviewer #1: Yes

Reviewer #2: Yes

5. Is the manuscript presented in an intelligible fashion and written in standard English?

Reviewer #1: Yes

Reviewer #2: Yes

6. Review Comments to the Author

Reviewer #1: (No Response)

Reviewer #2: There is no more comment. There is no more comment.

There is no more comment.

There is no more comment.

7. PLOS authors have the option to publish the peer review history of their article (what does this mean?). If published, this will include your full peer review and any attached files.

Reviewer #1: No

Reviewer #2: No

---

## [Editor Report · Acceptance letter]

25 Mar 2024

PONE-D-23-34507R1 

PLOS ONE

Dear Dr. Abebe, 

I'm pleased to inform you that your manuscript has been deemed suitable for publication in PLOS ONE. Congratulations! Your manuscript is now being handed over to our production team.

Kind regards, 

on behalf of

Dr. Kahsu Gebrekidan 

Academic Editor

PLOS ONE